# Fine-tuning Diffusion Policies with Backpropagation Through Diffusion Timesteps

## Abstract

Diffusion policies, widely adopted in decision-making scenarios such as robotics, gaming and autonomous driving, are capable of learning diverse skills from demonstration data due to their high representation power. However, the sub-optimal and limited coverage of demonstration data could lead to diffusion policies that generate sub-optimal trajectories and even catastrophic failures. While reinforcement learning (RL)-based fine-tuning has emerged as a promising solution to address these limitations, existing approaches struggle to effectively adapt Proximal Policy Optimization (PPO) to diffusion models. This challenge stems from the computational intractability of action likelihood estimation during the denoising process, which leads to complicated optimization objectives. In our experiments starting from randomly initialized policies, we find that online tuning of Diffusion Policies demonstrates much lower sample efficiency compared to directly applying PPO on MLP policies (MLP+PPO). To address these challenges, we introduce NCDPO, a novel framework that reformulates Diffusion Policy as a noise-conditioned deterministic policy. By treating each denoising step as a differentiable transformation conditioned on pre-sampled noise, NCDPO enables tractable likelihood evaluation and gradient backpropagation through all diffusion timesteps. Our experiments demonstrate that NCDPO achieves sample efficiency comparable to MLP+PPO when training from scratch, outperforming existing methods in both sample efficiency and final performance across diverse benchmarks, including continuous robot control and multi-agent game scenarios. Furthermore, our experimental results show that our method is robust to the number denoising timesteps.

## 1 Introduction

Recently, diffusion models have been widely adopted as policy classes in decision-making scenarios such as robotics [5, 21, 2, 33, 27, 15], gaming [15, 34], and autonomous driving [13, 30]. Although Diffusion Policies have shown remarkable capabilities in learning diverse behaviors from demonstration data [5], Diffusion Policy could show sub-optimal performance when the demonstration data is sub-optimal or only covers a limited set of environment states. To further optimize the performance of pretrained policies, Reinforcement Learning (RL) is adopted as a natural choice for fine-tuning pre-trained Diffusion Policies through interaction with the environment.

Currently, the most effective approach, DPPO (*Diffusion Policy Policy Optimization*) [20] employs Policy Gradient (PG) approaches to enhance the performance of pre-trained Diffusion Policy in continuous control tasks. By treating the denoising process of Diffusion Policy as a low-level Markov Decision Process, DPPO optimizes the Gaussian likelihood of all denoising steps. However, through our extensive experiments, we find fine-tuning Diffusion Policies with RL faces a challenge of sample efficiency. Specifically, in our RL experiments starting from randomly initialized policies, we find

Submitted to 39th Conference on Neural Information Processing Systems (NeurIPS 2025). Do not distribute.

that training Diffusion Policy with DPPO could lead to worse sample efficiency and final performance than training an MLP policy with standard RL. We hypothesize that the training efficiency gap occurs since DPPO uses a much larger action space for RL training, which impedes the sample efficiency of RL training. Therefore, a question becomes particularly important: *Can we design a more effective fine-tuning approach for Diffusion Policy that avoids enlarging the action space during RL training?*

In this work, we present *Noise-Conditioned Diffusion Policy Optimization* (NCDPO), a sample-efficient RL algorithm for fine-tuning Diffusion Policies. NCDPO formulates the denoising process of Diffusion Policy as a noise-conditioned inference process, ensuring the RL objective only contain the likelihood of the interactive actions, i.e. actions generated by Diffusion Policy to interact with the environment. In the policy update phase, the gradients with respect to the policy parameters are computed with *Backpropagation through Diffusion Timesteps* (BPDT). When performing RL training on randomly initialized policies, we show that training Diffusion Policy with NCDPO achieves comparable sample efficiency with training an MLP policy with RL.

In summary, our main contribution is NCDPO, a novel framework which is applicable to both continuous and discrete environments, to fine-tune Diffusion Policies, by formulating denoising steps as deterministic generation process and apply PPO. We also evaluate NCDPO on a set of environments, ranging from continuous robot control and multi-agent coordination tasks. We demonstrate that NCDPO obtains higher sample efficiency and stronger final performance than baseline methods across all evaluated environments. Finally, our ablation study reveals that that NCDPO is robust to the number of diffusion timesteps and remains highly sample efficient when the number of diffusion timesteps is large.

## 2  Related Work

**Diffusion Models and Diffusion Policies.** Diffusion-based generative models have demonstrated remarkable effectiveness in the domains of visual content generation [23, 26, 19].One central capability of Diffusion Models is the denoising process that iteratively refines sampled noises into clean datapoints. [9, 24, 25]. Beyond their success in content generation, diffusion models have increasingly been adapted for decision-making tasks across a range of domains, including robotics [5, 21, 2, 33, 27, 15], autonomous driving [13, 30], and gaming [15, 34]. In robotics, most existing work trains Diffusion Policies through imitation learning. For instance, Reuss et al. [21] predict future action chunks using goal-conditioned imitation learning, while [33] integrate Diffusion Policies with compact 3D representations extracted from point clouds. To further enhance the quality of generated behaviors, return signal or goal conditioning is applied to encourage the generation of high-value actions [10, 1, 12].

**Fine-tuning Diffusion Policy with Reinforcement Learning.** Recent works have aimed to enhance learned Diffusion Policy through fine-tuning with Reinforcement Learning approaches. A line of work has been focusing on integrating Diffusion Policies with Q-learning using offline data [4, 11, 28, 7, 22, 35, 18]. In addition to offline reinforcement learning, recent advancements have explored fine-tuning Diffusion Policies with online RL algorithms, for example, aligning the score function with the action gradient [31], or employing the diffusion model as a policy extraction mechanism within implicit Q-learning [8]. Most recently, [20] formulates the denoising process of Diffusion Policy as a "Diffusion MDP", enabling the application of RL algorithms to optimize all denoising steps with online feedback. In this work, we investigate an alternative representation for the denoising process that enables sample efficient fine-tuning of Diffusion Policy.

## 3  Preliminary

**Markov Decision Process.** A Markov Decision Process (MDP) is defined as a tuple $\mathcal{M} = \langle \mathcal{S}, \mathcal{A}, P_0, P, R, \gamma \rangle$ where $\mathcal{S}$ denotes the state space, $\mathcal{A}$ is the action space, $P_0$ is the distribution of initial states, $P$ is the transition function, $R$ is the reward function and $\gamma$ is the discount factor. At timestep $t$, a policy $\pi$ generates an action $a_t \in \mathcal{A}$ at state $s_t$. The goal is to find a policy $\pi$ that maximizes the objective of expected discounted return,

$$J(\pi) = \mathbb{E}_{s_t, a_t}[\sum_{t \geq 0} \gamma^t R(s_t, a_t)] \tag{1}$$

**Proximal Policy Optimization (PPO).** PPO is a reinforcement learning approach that optimizes the policy by estimating the policy gradient. In each iteration, given the last iteration policy $\pi_{\theta_k}$, PPO maximizes the clipped objective,

$$L(\theta|\theta_k) = \mathbb{E}_\tau \left[ \sum_t \min \left( \frac{\pi_\theta(a_t|s_t)}{\pi_{\theta_k}(a_t|s_t)} A^{\pi_{\theta_k}}(s_t, a_t), \right. \right.$$
$$\left. \left. \text{clip}\left( \frac{\pi_\theta(a_t|s_t)}{\pi_{\theta_k}(a_t|s_t)}, 1 - \epsilon, 1 + \epsilon \right) A^{\pi_{\theta_k}}(s_t, a_t) \right) \right] \tag{2}$$

where $A^{\pi_{\theta_k}}(s_t, a_t)$ is the estimated advantage for action $a_t$ at state $s_t$.

**Diffusion Policy.** Diffusion Policy $\pi_\theta$ is a diffusion model that generates actions $a$ by conditioning on states $s$. In Diffusion Policy training, the *forward process* gradually adds Gaussian noise to the training data to obtain a chain of noisy datapoints $a^0, a^1, \ldots, a^K$,

$$q(a^{1:K}|a^0) := \prod_{k=1}^K q(a^k|a^{k-1}), \qquad q(a^k|a^{k-1}) := \mathcal{N}(a^k; \sqrt{1 - \beta_k}a^{k-1}, \beta_k I) \tag{3}$$

Diffusion Policy could generate actions with a *reverse process* or *denoising process* that gradually denoises a Gaussian noise $a^K \sim \mathcal{N}(a^K; 0, I)$ with learned Gaussian transitions,

$$\pi_\theta(a^{0:K}|s) := \prod_{k=1}^K \pi_\theta(a^{k-1}|a^k, s), \qquad \pi_\theta(a^{k-1}|a^k, s) := \mathcal{N}(a^{k-1}|\mu_\theta(a^k, k, s), \sigma_k^2 I) \tag{4}$$

where $\sigma$ is a fixed noise schedule for action generation, $\beta$ denotes the forward process variances and is held as constant, and $\theta$ is the parameter of Diffusion Policy. To avoid ambiguity, we use *interactive actions* to denote the action $a^0$ that is used for interacting with the environment and *latent actions* to denote actions $a^1, \cdots, a^K$ that are generated during the denoising process. For more training details on diffusion models, please refer to [9].

**Diffusion Policy Policy Optimization (DPPO).** Note that the action likelihood $\pi_\theta(a_t^0|s_t)$ of Diffusion Policy $\pi_\theta$ is intractable,

$$\pi_\theta(a_t^0|s_t) = \int_{a_t^1, \cdots, a_t^K} \mathbb{P}[a_t^0, \cdots, a_t^K|s_t, \pi_\theta] \cdot da_t^1 \cdots da_t^K$$

The intractability of the action likelihood makes it impossible to directly fine-tune Diffusion Policy with PPO since the RL loss (Eq. 2) requires computing the exact action likelihood. To address this challenge, DPPO [20] proposes to formulate the denoising process as a low-level "Diffusion MDP" $\mathcal{M}_{\text{Diff}}$. In $\mathcal{M}_{\text{Diff}}$, a state is defined as a combination of the environment state and a *latent action* $\hat{s}_t^k = (s_t, a_t^k)$. For $k = K, \cdots, 1$, the transition from $\hat{s}_t^k$ to $\hat{s}_t^{k-1}$ represents the denoising process and takes no actual change on the environment state. For a denoising step $k \in [1, K]$, the state $\hat{s}_t^k = (s_t, a_t^k)$ transits to $\hat{s}_t^{k-1} = (s_t, a_t^{k-1})$. After the denoising process is finished at $k = 0$, the interactive action $a_t^0$ is used to interact with the environment and triggers the environment transition, i.e. the next state would be $\hat{s}_{t+1}^K = (s_{t+1}, a_{t+1}^K)$ where $s_{t+1} \sim P(s_t, a_t^0)$ and $a_{t+1}^K \sim N(0, I)$ is a newly generated Gaussian noise.

## 4 Sample Efficiency Challenge for Diffusion Policy Fine-Tuning

In this section, we aim to investigate the sample efficiency of fine-tuning Diffusion Policy with RL. Specifically, we compare the sample efficiency of training Diffusion Policy using DPPO and training an MLP policy using standard PPO. Since our study only focuses on the RL process, both the MLP policy and Diffusion Policy are randomly initialized before RL training without performing any additional behavior cloning. For conciseness, we denote training Diffusion Policy with DPPO as *DP+DPPO* and training an MLP policy with PPO as *MLP+PPO*.

Our investigations are carried out on two OpenAI Gym locomotion tasks, Walker2D and Halfcheetah. The training curves of MLP+PPO and DP+DPPO are shown in Fig. 1. Although Diffusion Policy has

more powerful representation power than MLP policy [5], our results here surprisingly show that DP+DPPO is less sample efficient than MLP+PPO and could only achieves sub-optimal performance.

Why does this efficiency gap occur? We hypothesize that the underlying reason is that, by employing a two-level MDP formulation, DPPO actually significantly lengthens the MDP horizons in RL training to contain both the interactive actions and latent actions. This lengthened MDP horizon then results in difficulty in proper credit assignment. This insight raises a critical question: *Can we design an alternative RL algorithm for Diffusion Policy fine-tuning that avoids enlarging the action space?*

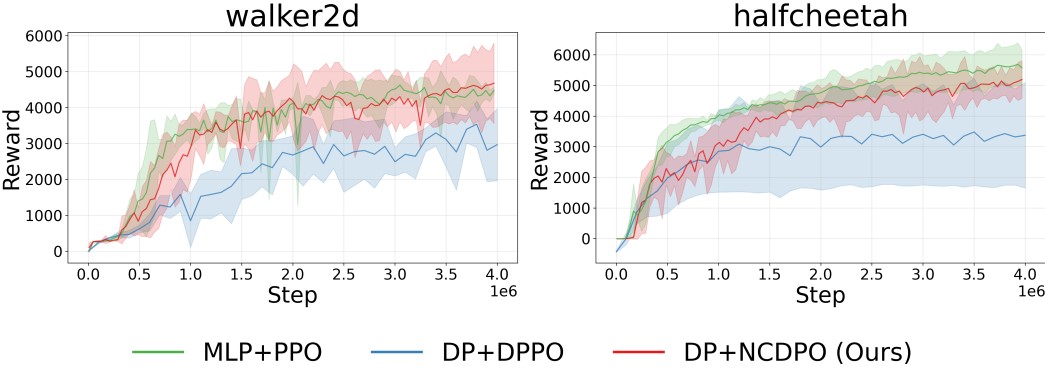

Figure 1: RL training from randomly initialized policy on Walker2D and HalfCheetah. Results are averaged over three seeds. Training curves indicate that DP+DPPO is less sample efficient than MLP+PPO and only achieves sub-optimal performance. Our approach, NCDPO, could fine-tune Diffusion Policy with high sample efficiency.

# 5 Noise-Conditioned Diffusion Policy Optimization

As discussed in Sec. 4, fine-tuning Diffusion Policy with a two-level MDP formulation could lead to sub-optimal sample efficiency and final performance. In this section, we present a novel sample-efficient RL training method for Diffusion Policy, *Noise-Conditioned Diffusion Policy Optimization (NCDPO)*. In Sec. 5.1, we show that NCDPO formulates the denoising process of Diffusion Policy as a noise-conditioned inference process. In Sec. 5.2, we show that NCDPO ensures PPO training operates on the same action space as the environment, without relying on optimizing action likelihood of latent actions.

## 5.1 Denoising Process as a Noise-Conditioned Inference Process

**Noise-conditioned Action Generation.** We decouple the stochastic and deterministic components of the denoising process. The stochastic component encompasses all the random noises sampled during the denoising process. The deterministic component further operates on these sampled noises with the model $\mu_\theta$.

Formally, Eq. 4 can be equivalently represented as,

$$a^{k-1} = \mu_\theta(a^k, k, s) + \sigma_k \cdot z^k \quad \text{where } z^k \sim \mathcal{N}(0, I) \tag{5}$$

A straitforward indication of Eq. 5 is that, in each denoising step, the only stochastic component is the Gaussian noise $z^k$, while the computation of $\mu_\theta(a^k, k, s)$ and addition between $\mu_\theta(a^k, k, s)$ and $\sigma_k \cdot z^k$ are both deterministic. Therefore, the whole denoising process can be split into a noise sampling phase and a deterministic inference phase.

In the *noise sampling phase*, we generate a sequence of standard Gaussian noises $z^1, \cdots, z^K$,

$$z^k \sim \mathcal{N}(0, I) \quad \text{for } k = K, K-1, \ldots, 1 \tag{6}$$

$$a^K \sim \mathcal{N}(0, I) \tag{7}$$

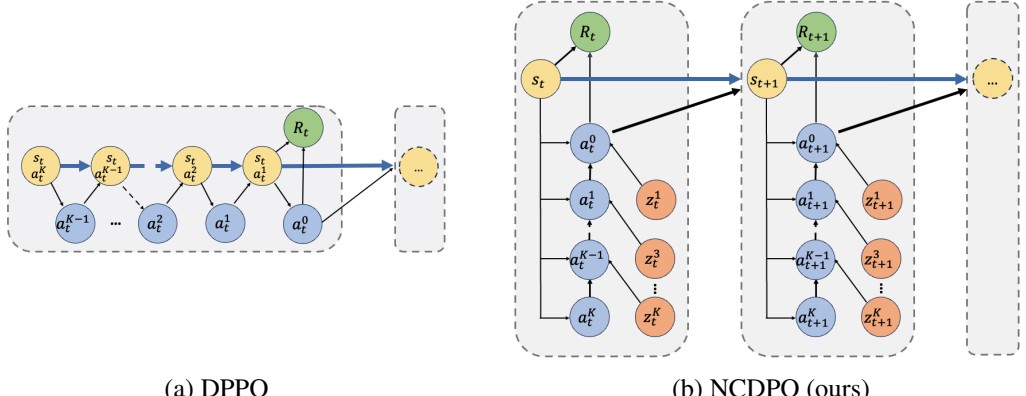

|  (a) DPPO | (b) NCDPO (ours) |

Figure 2: DPPO adopts a two-layer MDP design by combining the environment state with latent actions to form augmented states. In contrast, in each step, NCDPO first samples a group of random noises and computes the action based on the noises, resulting in a deterministic generation process (Eq. 9). Blue arrows in the figure indicate MDP transitions.

In the *deterministic inference phase*, given noises $z^1, \cdots, z^K$, $\mu_\theta$ is be used to compute the latent actions $a^k$ one by one. For $k = K, K-1, \ldots, 1$, $a^{k-1}$ is a linear combination of $\mu_\theta(a^k, k, s)$ and $z^k$,

$$a^{k-1} = \mu_\theta(a^k, k, s) + \sigma_k \cdot z^k \tag{8}$$

Consequently, the generated action $a^0$ can be computed by recursively applying Eq. 8,

$$
\begin{aligned}
a^0 &= \mu_\theta(\mu_\theta(\ldots \mu_\theta(a^K, K, s) \ldots, 2, s) + \sigma_2 \cdot z^2, 1, s) + \sigma_1 \cdot z^1 \\
&= f_\theta(s, a^K, z^{1:K})
\end{aligned} \tag{9}
$$

**MDP with Noise-augmented States.** As derived in Eq. 8 and Eq. 9, the denoising process can be partitioned into a noise sampling phase and a policy inference phase. We can incorporate the sampled noises into the MDP as part of the environment state. Formally, for the original environment MDP $\mathcal{M} = \langle \mathcal{S}, \mathcal{A}, P_0, P, R, \gamma \rangle$ we introduce *MDP with noise-augmented states* $\mathcal{M}_{noise} = \langle \mathcal{S}_{noise}, \mathcal{A}, P_0, P_{noise}, R, \gamma \rangle$. In $\mathcal{M}_{noise}$, each state $s_{noise}$ consists of a environment state $s \in \mathcal{S}$ and Gaussian noises $a^K, z^1, \cdots, z^K$. $\mathcal{M}_{noise}$ shares the same action space and reward function as the original MDP $\mathcal{M}$. In each decision-making step, the environment state transits to a new one, and the noises are all re-sampled.

**Denoising Process as a Noise-Conditioned Policy.** Given a noise-augmented state $s_{noise} = (s, a^K, z^{1:K})$, the deterministic inference phase of the denoising process can be represented as a *Noise-Conditioned Policy* $\pi_\theta^{NC}$ that generates the action $a^0$ using Eq. 9

Note that this noise-conditoned policy is a deterministic policy and could not be directly trained with PPO since the policy loss (Eq. 2) relies on a stochastic policy. Therefore, we introduce an additional operation to transform this deterministic policy into a stochastic one. Specifically, for continuous action space, we sample the final action $a^0$ near $f(a^K, z^{1:K})$ from a Gaussian distribution with a learnable standard variation $\sigma_{act}$,

$$\pi_\theta^{NC}(\cdot | z^{1:K}, a^K, s) = \mathcal{N}(f_\theta(s, a^K, z^{1:K}), \sigma_{act}^2) \tag{10}$$

For discrete action space, we use softmax to sample the final action by treating $f_\theta(s, a^K, z^{1:K})$ as logits,

$$\pi_\theta^{NC}(a^0 = i | z^{1:K}, a^K, s) \propto \exp(f_\theta(s, a^K, z^{1:K})_i / T) \tag{11}$$

where $i$ is the action index. $T$ is the temperature that allows the policy network to produce sharper action distributions.

 **5.2   Finetuning Noised-Conditioned Policy with PPO**

Under the formulation of NCDPO, at each denoising timestep $t$, Gaussian noise $z^t$ is first sampled, and the action is then generated via Eq. 10 or Eq. 11. This way, we can apply PPO objective in Eq.12 , which utilizes a clipped objective to regularize updated policy from original policy, to optimize interactive action probabilities:

$$
L(\theta|\theta_k) = \mathbb{E}_{a_t^0 \sim \pi_{\theta_k}^{NC}(z_t^{1:K}, a_t^K, s_t)} \left[ \sum_t \min\left( \frac{\pi_\theta^{NC}(a_t^0|z_t^{1:K}, a_t^K, s_t)}{\pi_{\theta_k}^{NC}(a_t^0|z_t^{1:K}, a_t^K, s_t)} A^{\pi_{\theta_k}^{NC}}(a_t^0|s_t), \right.\right.
$$
$$
\left.\left. \text{clip}\left( \frac{\pi_\theta^{NC}(a_t^0|z_t^{1:K}, a_t^K, s_t)}{\pi_{\theta_k}^{NC}(a_t^0|z_t^{1:K}, a_t^K, s_t)}, 1 - \epsilon, 1 + \epsilon \right) A^{\pi_{\theta_k}^{NC}}(a_t^0|s_t) \right) \right] \tag{12}
$$

As illustrated in Figure 1, in policy rollout process, each step begins by sampling a sequence of noises, which are then used by the Diffusion Policy to generate the corresponding action. These sampled noises are stored in the buffer. During training phase, the stored noises are reused to recompute the actions, enabling gradient backpropagation through the entire denoising process. This allows PPO to directly update all denoising steps of the diffusion policy.

---

**Algorithm 1** NCDPO

---

**Require:** Noise-conditinoed policy $\pi_\theta^{NC}$, noise scheduler $\sigma$
 1: **Parameters:** $\gamma \in [0, 1)$, $\varepsilon \in (0, 1)$, $N_{\text{episodes}}$, $N_{\text{PPO}}$
 2: **for** $e = 1, 2, \ldots, N_{\text{episodes}}$ **do**
 3:     buffer $\leftarrow \emptyset$
 4:     **for** $t = 0, 1, 2, \ldots, T - 1$ **do**
 5:         $a_t^K, z_t^{1:K} \sim \mathcal{N}(0, I)$
 6:         Sample $a_t^0$ from $\pi^{NC}(\cdot|z_t^{1:K}, a_t^K, s_t)$
 7:         $\log \pi_t^{NC} \leftarrow \pi^{NC}(a_t^0|z_t^{1:K}, a_t^K, s_t)$
 8:         Execute $a_t$, observe $r_t$, $s_{t+1}$
 9:         buffer $\leftarrow$ buffer $\cup \{s_t, a_t^K, r_t, \log \pi_t, z_t^{1:K}\}$
10:     **end for**
11:     **for** epoch $= 1, 2, \ldots, N_{\text{PPO}}$ **do**
12:         **for** mini-batch $b = 1, 2, \ldots$ **do**
13:             Calculate PPO loss $L(\theta|\theta_k)$ in Eq. 12, backpropagate gradients through diffusion timesteps and update parameter $\theta$
14:         **end for**
15:     **end for**
16: **end for**

---

As Fig.2 shows, NCDPO models the denoising process as deterministic generation conditioned on pre-sampled noise $z_t^{1:K}$. During inference, interactive actions are obtained through recursive model inference in Eq. 9 and applying the action sampling step in Eq. 10 and Eq. 11.

## 6   Experiments

In this section, we provide a comprehensive evaluation of NCDPO across a variety of challenging environments. We begin by detailing the experimental setup in Sec. 6.1, followed by results on continuous robot control tasks in Sec. 6.2 and discrete multi-agent coordination tasks in Sec.6.3. Finally, we conduct ablation studies in Sec. 6.4 to assess the robustness and of NCDPO.

### 6.1   Environmental Setup

**Environments: OpenAI Gym locomotion.**   Our first set of experiments involves testing NCDPO on a series of well-established locomotion benchmarks from OpenAI Gym [3], namely: `Hopper-v2`, `Walker2D-v2`, and `HalfCheetah-v2`. The pre-trained Diffusion Policies used in these experiments

are trained from the D4RL "medium" dataset [6], which contains a diverse range of pre-recorded trajectories. For the fine-tuning process, we use **dense** reward.

**Environments: Robomimic.**   We further evaluate the performance of NCDPO on robotic manipulation tasks within Robomimic benchmarks [14]. The specific scenarios we consider include `Lift`, `Can`, `Square`, and `Transport`, varying in difficulty. To ensure temporal consistency in actions, we employ action chunking with size 4 for `Lift`, `Can`, and `Square`, and size 8 for `Transport`, following the setting in [20]. All tasks are fine-tuned using **sparse** rewards, which provide feedback only in the form of success or failure signals.

**Environments: Google Research Football.** To evaluate NCDPO's capability in large discrete action spaces, we test on three Google Research Football scenarios requiring multi-agent coordination: `3 vs 1 with Keeper`, `Counterattack Hard`, and `Corner`. Here we adopt a centralized control strategy, where actions for all agents are generated simultaneously using a single Diffusion Policy. The base Diffusion Policies are pre-trained to output one-hot vectors corresponding to ground-truth actions. To construct the pre-training dataset, we aggregate trajectories from multiple MLP-based policies with varying success rates.

## 6.2   Evaluation on Continuous Robot Control Tasks

We first evaluate NCDPO on continuous control tasks across two benchmarks: OpenAI Gym locomotion and Robomimic. In these environments, we compare **NCDPO** with **DPPO** [20]; **DRWR** and **DAWR** [20], based on reward-weighted regression [17] and advantage-weighted regression respectively [16]; **DIPO** [31], which employs action gradients as the score function for denoising steps; and Q-learning-based methods such as **IDQL** [8] and **DQL** [28].

From the experimental results shown in Fig. 3 and Fig. 4, we observe that NCDPO consistently achieves the strongest performance and exhibits robustness across all tasks. While DPPO, which is the best among the baseline methods, performs comparably to NCDPO in the Robomimic benchmark [1], it lags behind in the OpenAI Gym locomotion environments. Other baselines generally underperform relative to DPPO. Notably, IDQL demonstrates strong performance on the first three Robomimic tasks but fails in the final one. In contrast, DQL suffers from instability across all scenarios. We additionally conducted experiments on `Square` using vision-based inputs, with results provided in the appendix. The results demonstrate consistent improvements in Diffusion Policy performance when fine-tuned with NCDPO.

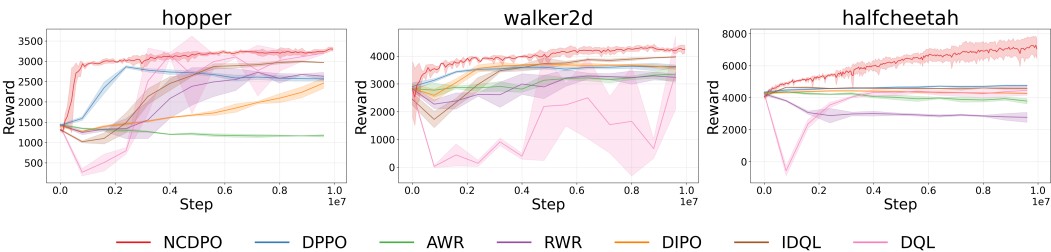

Figure 3: Performance comparison on OpenAI Gym locomotion tasks. Results are averaged over three seeds. NCDPO (ours) achieves the strongest performance.

---

[1]Note that we tested performance using the latest version of DPPO (v0.8), which is about 2.5x sample efficient in `Transport` task as reported in the original paper.

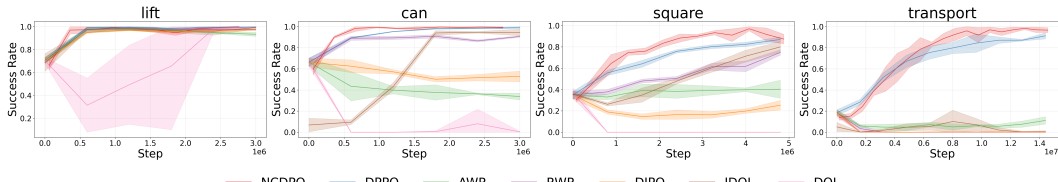

Figure 4: Performance comparison on Robomimic tasks. Results are averaged over three seeds. NCDPO (ours) achieves the strongest performance.

| Scenario | NCDPO (Ours) | DPPO | AWR | IDQL | DQL | RWR | DIPO |
|---|---|---|---|---|---|---|---|
| Hopper-Medium | **126.4** (1.8) | 98.4 (2.0) | 44.8 (1.2) | 113.8 (0.2) | 122.7 (1.2) | 100.9 (3.6) | 94.4 (4.9) |
| Hopper-Medium-Replay | **128.2** (2.8) | 114.6 (3.3) | 82.1 (7.0) | 117.9 (1.1) | 121.1 (3.6) | 104.2 (3.1) | 108.6 (1.0) |
| Hopper-Medium-Expert | **135.2** (2.0) | 102.4 (5.2) | 40.7 (0.5) | 131.9 (0.4) | 120.9 (28.1) | 113.6 (4.7) | 127.5 (1.3) |
| Walker2d-Medium | **118.4** (3.8) | 101.2 (1.6) | 93.5 (8.3) | 110.7 (0.5) | 94.9 (36.9) | 90.2 (3.3) | 99.8 (2.0) |
| Walker2d-Medium-Replay | **126.6** (4.5) | 105.1 (4.3) | 75.8 (2.3) | 121.9 (2.6) | 107.2 (37.8) | 69.2 (6.0) | 88.6 (11.0) |
| Walker2d-Medium-Expert | **141.0** (2.3) | 137.5 (2.0) | 124.2 (5.4) | 135.5 (2.7) | 67.3 (47.6) | 106.8 (6.8) | 138.8 (1.4) |
| HalfCheetah-Medium | **122.0** (11.0) | 82.3 (0.7) | 65.5 (2.9) | 79.3 (0.8) | 77.1 (5.4) | 47.9 (5.4) | 73.9 (0.6) |
| HalfCheetah-Medium-Expert | **139.7** (6.8) | 80.6 (1.6) | 64.4 (2.3) | 77.8 (1.0) | 72.1 (8.0) | 38.4 (2.9) | 71.4 (0.2) |
| HalfCheetah-Medium-Replay | **121.0** (2.0) | 72.3 (0.4) | 60.5 (0.7) | 74.3 (0.9) | 73.0 (3.1) | 30.7 (1.8) | 58.1 (0.9) |
| Lift | **100.0** (0.0) | 99.7 (0.2) | 93.3 (1.7) | 99.2 (0.1) | 99.8 (0.3) | 97.5 (0.5) | 97.3 (0.8) |
| Can | **99.3** (1.2) | 99.0 (1.0) | 33.8 (3.2) | 94.5 (3.1) | 0.3 (0.6) | 90.7 (0.8) | 52.8 (5.1) |
| Square | **87.3** (4.5) | 87.0 (2.3) | 40.3 (8.5) | 80.0 (5.0) | 0.0 (0.0) | 74.8 (2.5) | 25.3 (4.5) |
| Transport | **96.7** (2.31) | 91.3 (2.9) | 11.2 (3.5) | 0.5 (0.8) | 0.0 (0.0) | 0.0 (0.0) | 0.2 (0.3) |

Table 1: Mean and standard deviation of performance over continuous robot control scenarios. Each result is evaluated on three different seeds. NCDPO (ours) exhibits the strongest performance. Performance on OpenAI Gym locomotion tasks are normalized according to scores of MLP policies trained from scratch using PPO with 1M samples reported in Tianshou [29] . Original scores are listed in Table 3.

## 6.3  Evaluation on Discrete Multi-agent Coordination Tasks

Following our evaluation on continuous control tasks, we next examine NCDPO on Google Research Football, a benchmark for cooperative multi-agent control. To facilitate more effective coordination among agents, we adopt a centralized multi-agent control strategy in which actions for all agents are generated simultaneously. This formulation leverages the high representational capacity of diffusion models to model complex inter-agent dependencies. However, it also gives rise to a high-dimensional joint action space (i.e., num_agents $\times$ actions), presenting substantial challenges for reinforcement learning fine-tuning.

We compare NCDPO with MLP policies trained using Multi-Agent Proximal Policy Optimization (MAPPO) [32]. This baseline is initialized through behavior cloning using a Cross-Entropy loss function. As no public dataset exists for Google Research Football, a custom dataset is constructed to pre-train Diffusion Policies by training multiple MAPPO agents [32] with different random seeds and early-stopping them at various stages. These agents exhibit varying winning rates and employ diverse tactical behaviors.

As Fig. 5 demonstrates, NCDPO outperforms the MLP baseline across all three evaluated scenarios. This outcome not only highlights the superiority of Diffusion Policy in handling complex and diverse demonstration data over simple MLP policy, but also confirms the effectiveness of the DP+NCDPO during the fine-tuning phase.

| Scenario | NCDPO (Ours) | MAPPO |
|---|---|---|
| 3 vs 1 with Keeper | **87.4**(2.7) | 75.1(12.3) |
| Counterattack Hard | **87.0**(2.5) | 80.0(2.0) |
| Corner | **78.3**(4.5) | 74.9(3.0) |

Table 2: Average evaluation success rate and standard deviation (over three seeds) on Google Research Football scenarios. The base Diffusion Policy and MLP policy are pre-trained on the same dataset. MLP policy is trained using Cross-Entropy loss.

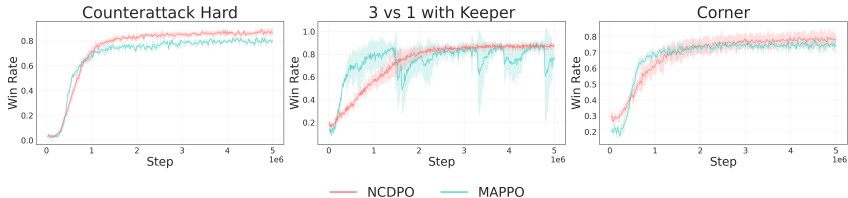

Figure 5: Performance comparison in Google Research Football. Results are averaged over at least three seeds. NCDPO (ours) exhibits strong performance and stability.

## 6.4 NCDPO is Robust to the Number of Denoising Steps

We further conduct an ablation study to investigate the impact of varying the number of denoising steps in the diffusion model. The experimental results shown in Fig. 7 indicate that NCDPO demonstrates strong robustness to the choice of denoising steps.

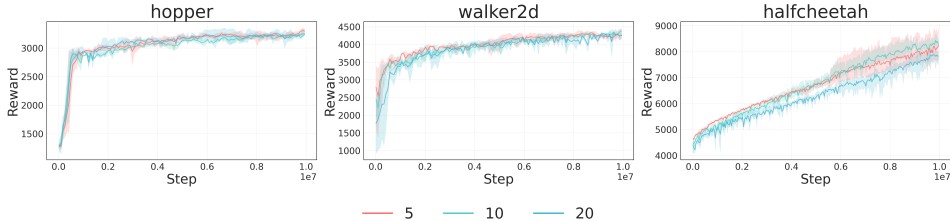

Figure 6: Ablation Study on Denoising Steps in OpenAI Gym locomotion tasks.

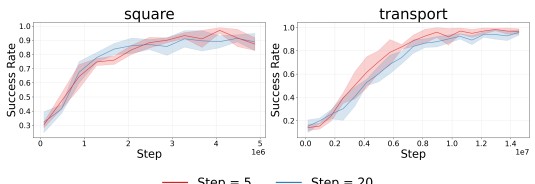

Figure 7: Ablation Study on Denoising Steps in Robomimic tasks.

We hypothesize that the robustness of NCDPO arises from the way gradients are propagated through time during the diffusion process. This gradient flow leads to more accurate gradient estimates.

## 7 Conclusion and Limitations

We present NCDPO, a novel approach for fine-tuning Diffusion Policies through Proximal Policy Optimization that exhibits strong performance across continuous and discrete control domains. Our key innovation lies in reformulating the diffusion denoising process as a noise-conditioned stochastic policy that enables effective gradient backpropagation through diffusion timesteps. Through extensive experiments across locomotion, manipulation, and multi-agent cooperation scenarios, we demonstrate that NCDPO achieves superior sample efficiency and final performance compared to existing diffusion RL approaches. NCDPO's ability to handle both continuous and discrete action spaces suggests its potential as a general-purpose policy optimization framework.

Our study focuses on the algorithmic development and evaluation of NCDPO in simulated settings. Consequently, we have not yet explored sim-to-real transfer on physical robots. These choices reflect our emphasis on fine-tuning methodology. Extending NCDPO to real-world deployment remains to be implemented in future work.

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

## A  Self-Imitation Regularizer

When directly fine-tuning Diffusion Policies using policy gradient methods, we observe a structure collapse issue—namely, the Diffusion Policy fails to maintain consistency between the forward and reverse processes. To preserve the structural integrity of the diffusion model, we introduce self-imitation regularization. Specifically, we perform behavior cloning on the trajectories generated in the previous episode. Empirically, we find that this regularization significantly reduces the behavior cloning loss. In contrast, without it, this behavior cloning loss will keep increasing, indicating structural degradation in the Diffusion Policy.

## B  Additional experimental results

### B.1  Original Scores on OpenAI Gym locomotion tasks

| Scenario | NCDPO (Ours) | DPPO | AWR | IDQL | DQL | RWR | DIPO | PPO |
|---|---|---|---|---|---|---|---|---|
| Hopper-Medium | **3297.3** (47.8) | 2566.6 (51.1) | 1168.9 (30.5) | 2970.2 (5.2) | 3200.7 (30.1) | 2633.6 (94.0) | 2463.1 (127.6) | 2609.3 |
| Hopper-Medium-Replay | **3345.64** (71.88) | 2988.97 (86.90) | 2142.30 (183.82) | 3076.91 (29.82) | 3159.72 (94.85) | 2718.27 (79.86) | 2834.07 (25.00) | 2609.3 |
| Hopper-Medium-Expert | **3528.74** (51.89) | 2672.64 (135.15) | 1062.47 (13.15) | 3440.50 (10.79) | 3153.66 (733.36) | 2964.11 (121.35) | 3326.99 (32.73) | 2609.3 |
| Walker2d-Medium | **4248.8** (137.6) | 3632.1 (55.9) | 3353.9 (296.9) | 3972.8 (17.3) | 3405.2 (1322.7) | 3238.3 (116.8) | 3581.6 (70.3) | 3588.5 |
| Walker2d-Medium-Replay | **4544.59** (162.98) | 3770.52 (154.50) | 2719.04 (83.84) | 4373.89 (91.66) | 3846.26 (1357.97) | 2483.04 (215.70) | 3180.80 (396.27) | 3588.5 |
| Walker2d-Medium-Expert | **5060.92** (82.87) | 4935.57 (73.28) | 4458.68 (195.26) | 4863.91 (95.34) | 2416.68 (1708.34) | 3831.65 (243.76) | 4979.96 (50.22) | 3588.5 |
| HalfCheetah | **7058.8** (635.1) | 4758.3 (41.8) | 3788.7 (166.5) | 4584.4 (45.3) | 4459.1 (309.5) | 2773.1 (310.0) | 4272.4 (33.2) | 5783.9 |
| HalfCheetah-Expert | **8079.30** (392.21) | 4663.23 (92.62) | 3723.21 (131.37) | 4499.39 (57.61) | 4171.97 (465.25) | 2218.39 (168.36) | 4126.96 (12.14) | 5783.9 |
| HalfCheetah-Replay | **7000.14** (113.88) | 4181.57 (24.59) | 3501.08 (40.34) | 4295.81 (53.02) | 4223.07 (177.28) | 1775.98 (101.28) | 3362.48 (49.47) | 5783.9 |

Table 3: Mean and standard deviation of original scores across continuous robot control scenarios.

### B.2  Experiments with Vision Inputs

We performed evaluation on `Square` task in robomimic with vision as input. Results demonstrate the effectiveness of NCDPO.

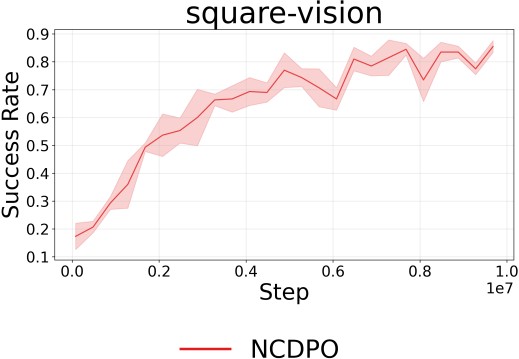

Figure 8: Experimental results for vision inputs.

### B.3  Ablation Study

We observe that setting the action chunk size to one significantly improves performance in Gym environments. We hypothesize that this is due to the nature of these tasks, where agents must respond promptly to rapid and continuous changes in the environment. Smaller chunk sizes allow the policy to adapt its actions more frequently, which is crucial for achieving fine-grained control. The corresponding results are presented in Figure 9.

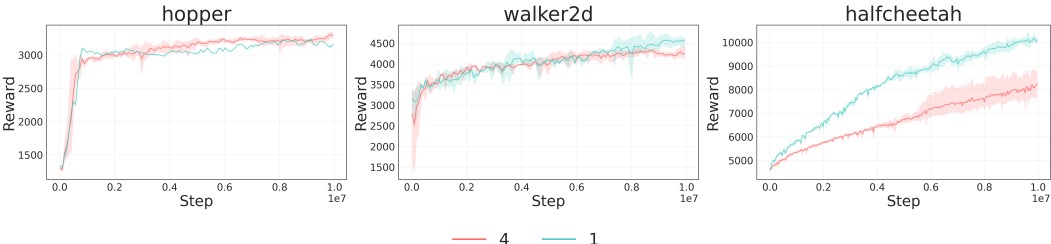

Figure 9: Ablation study on action chunk size in OpenAI Gym locomotion tasks.

additionally, in discrete action environments, modifying the noise scheduler to increase Gaussian noise during the denoising steps improves exploration without degrading overall performance, as shown in Figure 10. In discrete settings, the absolute values of the logits are less important than their relative magnitudes, which allows increased noise to encourage exploration while preserving policy effectiveness. To achieve this, we adjust the noise scheduler using parameters $\eta$ and $\beta_{\text{base}}$, increasing the noise level via the transformation:

$$\beta'_k = \beta_{base} \left( \frac{\beta_k}{\beta_{base}} \right)^{\eta}$$

where $\beta_k$ corresponds to the original noise schedule defined in Equation 3. In our implementation, we set $\beta_{base} = 0.7$.

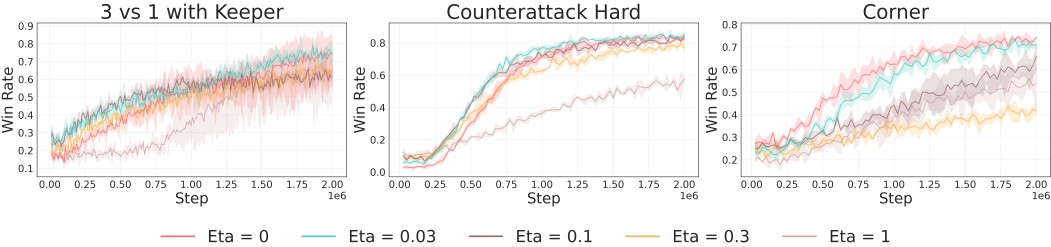

Figure 10: Ablation study on different values of $\eta$ in Google Research Football.

We further find that increasing the initial noise scale $\sigma_a$ in the acting layer enhances exploration. An ablation study conducted on Robomimic supports this finding:

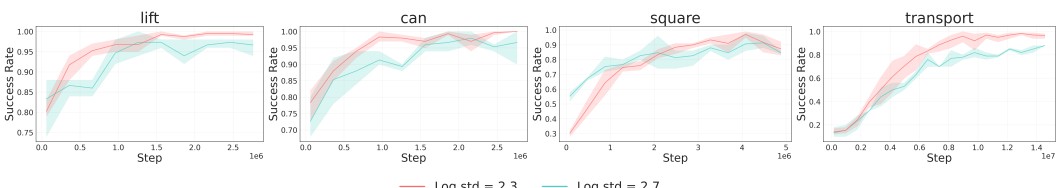

Figure 11: Ablation study of different choices of initial $\log \sigma_a$ in Robomimic tasks.

## B.4 Further Experiments in OpenAI Gym locomotion tasks.

We evaluated different training methods using datasets of varying quality for pretraining the base policy. The "medium-replay" dataset consists of replay buffer samples collected before early stopping, while the "medium-expert" dataset contains equal proportions of expert demonstrations and suboptimal rollouts [6].

Regardless of dataset quality, NCDPO consistently outperforms all baselines, as shown in Figures 12 and 13.

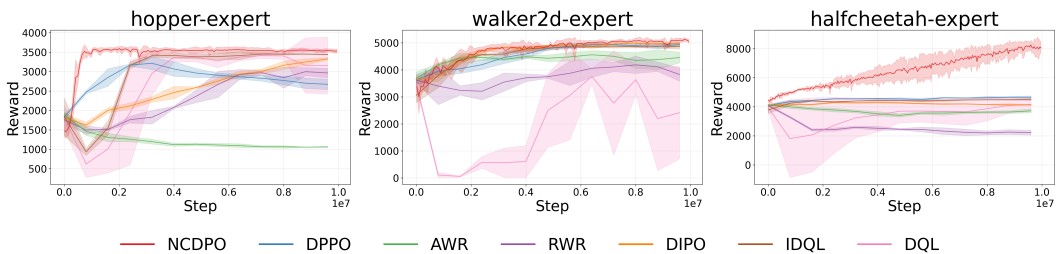

Figure 12: Pretraining with expert datasets.

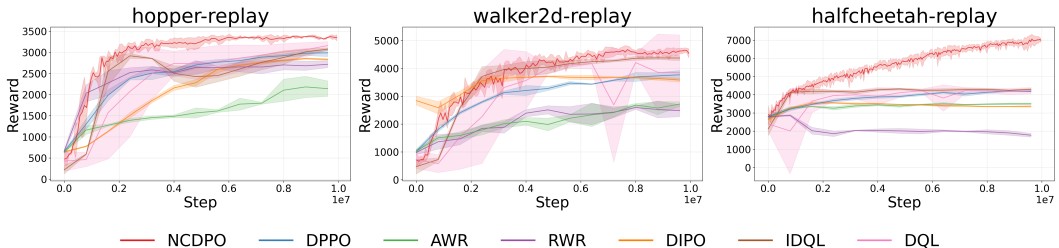

Figure 13: Pretraining with replay datasets.

### B.5 Google Research Football Data Curation

For each scenario, we collected 200K environment steps per model. The win rates of the agents used for dataset generation are summarized in Table 4.

| Scenario | Win Rates |
|---|---|
| 3 vs 1 with Keeper | 0.93, 0.90, 0.70, 0.55 |
| Corner | 0.76, 0.75, 0.50, 0.50, 0.41 |
| Counterattack Hard | 0.90, 0.78, 0.70, 0.61, 0.56, 0.56 |

Table 4: Win rates of trained agents used for dataset collection in Google Research Football. Each model contributes 200,000 steps.

## C Implementation Details and Hyperparameters

For NCDPO, we apply Adam optimizer for actor and AdamW optimizer for critic. For all other baselines, AdamW optimizer is adopted.

For fair comparison, we adopt the same network architecture as DPPO [20] and directly utilize their implementation for model structure. Our overall training framework is built upon a modified codebase of MAPPO [32].

| Task | $\gamma$ | $\lambda$ | Action Chunk | Actor LR | Critic LR | Actor MLP Size | Critic MLP Size | Actor MLP Layers | $\eta$ | Initial Noise Log Std | $1/T$ | Denoising Step | Clone Epochs | Clone LR | Episode Length | Mini-batch Number | Environment Max Steps | Parallel Environments |
|---|---|---|---|---|---|---|---|---|---|---|---|---|---|---|---|---|---|---|
| Hopper | 0.995 | 0.985 | 4 | 3e-5 | 1e-3 | 1024 | 256 | 7 | 0 | -2 | - | 5 | 8 | 1e-3 | 256 | 1 | 1000 | 32 |
| Walker2d | 0.995 | 0.985 | 4 | 3e-5 | 1e-3 | 1024 | 256 | 7 | 0 | -2 | - | 5 | 8 | 1e-3 | 500 | 1 | 1000 | 32 |
| HalfCheetah | 0.99 | 0.985 | 4 | 3e-5 | 1e-3 | 1024 | 256 | 7 | 0 | -2 | - | 5 | 3 | 1e-4 | 500 | 1 | 1000 | 32 |
| lift | 0.999 | 0.99 | 4 | 3e-5 | 1e-3 | 1024 | 256 | 7 | 0 | -2.3 | - | 5 | 2 | 1e-3 | 300 | 4 | 300 | 200 |
| can | 0.999 | 0.99 | 4 | 3e-5 | 1e-3 | 1024 | 256 | 7 | 0 | -2.3 | - | 5 | 60 | 3e-4 | 300 | 4 | 300 | 200 |
| square | 0.999 | 0.99 | 4 | 3e-5 | 1e-3 | 1024 | 256 | 7 | 0 | -2.3 | - | 5 | 200 | 2e-4 | 400 | 4 | 400 | 200 |
| square-vision | 0.999 | 0.99 | 4 | 3e-5 | 5e-4 | 1024 | 256 | 7 | 0 | -2.3 | - | 5 | 200 | 2e-4 | 400 | 4 | 400 | 200 |
| transport | 0.999 | 0.99 | 8 | 3e-5 | 1e-3 | 1024 | 256 | 7 | 0 | -2.3 | - | 5 | 321 | 8e-4 | 800 | 4 | 800 | 200 |
| 3 vs 1 with Keeper | 0.99 | 0.95 | 1 | 3e-5 | 1e-3 | 1024 | 256 | 7 | 0.03 | - | 20 | 5 | 10 | 1e-3 | 200 | 1 | - | 50 |
| Corner | 0.99 | 0.95 | 1 | 3e-5 | 1e-3 | 1024 | 256 | 7 | 0.03 | - | 20 | 5 | 10 | 1e-3 | 500 | 1 | - | 50 |
| Counterattack Hard | 0.99 | 0.95 | 1 | 3e-5 | 1e-3 | 1024 | 256 | 7 | 0.03 | - | 20 | 5 | 10 | 1e-3 | 500 | 1 | - | 50 |

Table 5: Hyperparameter settings for different tasks of NCDPO.

| Task | $\gamma$ | $\lambda$ | Action Chunk | Actor LR | Critic LR | Actor MLP Size | Critic MLP Size | Actor MLP Layers | Denoising Step | Episode Length | Mini-batch Size | Environment Max Steps | Parallel Environments |
|------|------|------|------|------|------|------|------|------|------|------|------|------|------|
| Hopper | 0.99 | 0.95 | 4 | 1e-4 | 1e-3 | 512 | 256 | 3 | 20 | 2000 | 50000 | 1000 | 40 |
| Walker2d | 0.99 | 0.95 | 4 | 1e-4 | 1e-3 | 512 | 256 | 3 | 20 | 2000 | 50000 | 1000 | 40 |
| HalfCheetah | 0.99 | 0.95 | 4 | 1e-4 | 1e-3 | 512 | 256 | 3 | 20 | 2000 | 50000 | 1000 | 40 |
| lift | 0.999 | 0.95 | 4 | 1e-4 | 5e-4 | 512 | 256 | 3 | 20 | 1200 | 7500 | 300 | 50 |
| can | 0.999 | 0.95 | 4 | 1e-4 | 5e-4 | 512 | 256 | 3 | 20 | 1200 | 7500 | 300 | 50 |
| square | 0.999 | 0.95 | 4 | 1e-4 | 5e-4 | 512 | 256 | 3 | 20 | 1600 | 10000 | 400 | 50 |
| transport | 0.999 | 0.95 | 8 | 1e-4 | 5e-4 | 512 | 256 | 3 | 20 | 3200 | 10000 | 800 | 50 |

Table 6: Hyperparameter settings for Baselines in robot control. Experiment is executed using DPPO [20] implementation and hyperparameters. Batch size for all baselines other than DPPO is 1000. For further details, please refer to DPPO paper [20].

| Task | $\gamma$ | $\lambda$ | Action Chunk | Actor LR | Critic LR | Actor MLP Size | Critic MLP Size | Actor MLP Layers | $\eta$ | Initial Noise Log Std | $1/T$ | Denoising Step | Clone Epochs | Clone LR |
|------|------|------|------|------|------|------|------|------|------|------|------|------|------|------|
| 3 vs 1 with Keeper | 0.99 | 0.95 | 1 | 5e-4 | 5e-4 | 256 | 256 | - | - | - | 20 | 5 | 8 | 1e-3 |
| Corner | 0.99 | 0.95 | 1 | 5e-4 | 5e-4 | 256 | 256 | - | - | - | 20 | 5 | 8 | 1e-3 |
| Counterattack Hard | 0.99 | 0.95 | 1 | 5e-4 | 5e-4 | 256 | 256 | - | - | - | 20 | 5 | 8 | 1e-3 |

Table 7: Hyperparameter of MLP on football. Experiment is run on MAPPO codebase and MLP architecture remains same as MAPPO, and does not use residual connection, thus rendering parameter MLP layers unusable.

| Task | $\gamma$ | $\lambda$ | Action Chunk | Actor LR | Critic LR | Actor MLP Size | Critic MLP Size | Actor MLP Layers | $\eta$ | Initial Log Std | $1/T$ | Denoising Step |
|------|------|------|------|------|------|------|------|------|------|------|------|------|
| Walker2d-NCDPO | 0.995 | 0.985 | 1 | 1e-4 | 1e-3 | 256 | 256 | 3 | 0 | -0.8 | - | 5 |
| HalfCheetah-NCDPO | 0.99 | 0.985 | 1 | 1e-4 | 1e-3 | 256 | 256 | 3 | 0 | -0.8 | - | 5 |
| Walker2d-MLP+PPO | 0.995 | 0.985 | 1 | 1e-4 | 1e-3 | 256 | 256 | 3 | - | -0.8 | - | - |
| HalfCheetah-MLP+PPO | 0.99 | 0.985 | 1 | 1e-4 | 1e-3 | 256 | 256 | 3 | - | -0.8 | - | - |
| Walker2d-DPPO | 0.99 | 0.985 | 1 | 1e-4 | 1e-3 | 512 | 256 | 3 | - | - | - | 10 |
| HalfCheetah-DPPO | 0.99 | 0.985 | 1 | 1e-4 | 1e-3 | 512 | 256 | 3 | - | - | - | 10 |

Table 8: Hyperparameters for training from scratch. In this experiment, MLP+PPO has exatcly the same architecture with MLP in diffusion's denoising process. Numbers of mini-batches and parallel environments are the same as Table 5.

| Task | $\gamma$ | $\lambda$ | Action Chunk | Actor LR | Critic LR | Actor MLP Size | Critic MLP Size | Actor MLP Layers | $\eta$ | Initial Log Std | $1/T$ | Denoising Step | Clone Epochs | Clone LR |
|------|------|------|------|------|------|------|------|------|------|------|------|------|------|------|
| Hopper | 0.995 | 0.985 | 4 | 3e-5 | 1e-3 | 1024 | 256 | 7 | 0 | -2 | - | 5/10/20 | 8 | 1e-3 |
| Walker2d | 0.995 | 0.985 | 4 | 3e-5 | 1e-3 | 1024 | 256 | 7 | 0 | -2 | - | 5/10/20 | 8 | 1e-3 |
| HalfCheetah | 0.99 | 0.985 | 4 | 3e-5 | 1e-3 | 1024 | 256 | 7 | 0 | -2 | - | 5/10/20 | 8 | 1e-3 |
| square | 0.995 | 0.985 | 4 | 3e-5 | 5e-4 | 1024 | 256 | 7 | 0 | -2.3 | - | 5/20 | 8 | 5e-4 |
| transport | 0.99 | 0.985 | 8 | 3e-5 | 5e-4 | 1024 | 256 | 7 | 0 | -2.3 | - | 5/20 | 8 | 5e-4 |

Table 9: Hyperparameter of Ablation on Denoising Steps.

| Task | $\gamma$ | $\lambda$ | Action Chunk | Actor LR | Critic LR | Actor MLP Size | Critic MLP Size | Actor MLP Layers | $\eta$ | Initial Log Std | $1/T$ | Denoising Step | Clone Epochs | Clone LR |
|------|------|------|------|------|------|------|------|------|------|------|------|------|------|------|
| lfit | 0.999 | 0.99 | 4 | 3e-5 | 1e-3 | 1024 | 256 | 7 | 0 | -2.3/-2.7 | - | 5 | 2 | 1e-3 |
| can | 0.999 | 0.99 | 4 | 3e-5 | 1e-3 | 1024 | 256 | 7 | 0 | -2.3/-2.7 | - | 5 | 60 | 3e-4 |
| square | 0.999 | 0.99 | 4 | 3e-5 | 1e-3 | 1024 | 256 | 7 | 0 | -2.3/-2.7 | - | 5 | 200 | 2e-4/5e-4 |
| transport | 0.999 | 0.99 | 8 | 3e-5 | 1e-3 | 1024 | 256 | 7 | 0 | -2.3/-2.7 | - | 5 | 321 | 8e-4/5e-4 |

Table 10: Hyperparameter of Ablation on Initial Noise.

| Task | $\gamma$ | $\lambda$ | Action Chunk | Actor LR | Critic LR | Actor MLP Size | Critic MLP Size | Actor MLP Layers | $\eta$ | Initial Log Std | $1/T$ | Denoising Step | Clone Epochs | Clone LR |
|------|------|------|------|------|------|------|------|------|------|------|------|------|------|------|
| 3 vs 1 with Keeper | 0.99 | 0.95 | 1 | 3e-5 | 1e-3 | 1024 | 256 | 7 | 0.03/0.1/0.3/1 | - | 20 | 5 | 8 | 1e-3 |
| Corner | 0.99 | 0.95 | 1 | 3e-5 | 1e-3 | 1024 | 256 | 7 | 0.03/0.1/0.3/1 | - | 20 | 5 | 8 | 1e-3 |
| Counterattack Hard | 0.99 | 0.95 | 1 | 3e-5 | 1e-3 | 1024 | 256 | 7 | 0.03/0.1/0.3/1 | - | 20 | 5 | 8 | 1e-3 |

Table 11: Hyperparameter of Ablation on $\eta$. $\eta = 1$ indicates using original scheduler.

## D Computational Resources

Each run could be done in 6 hours with 1 AMD Ryzen 3990X 64-Core Processor and 1 NVIDIA 3090 GPU.

