# OpenReview forum: "Fine-tuning Diffusion Policies with Backpropagation Through Diffusion Timesteps"
_NeurIPS.cc/2025/Conference — Submitted to NeurIPS 2025_

### Official Review · Reviewer_ohxP · 2025-07-03

**Clarity:** 4
**Significance:** 3
**Originality:** 2
**Rating:** 4
**Confidence:** 4

**Summary:**

This paper proposes NCDPO, an improvement over DPPO for online diffusion policy optimization. DPPO is an RL algorithm which employs diffusion models to parameterize the policy and formulates the optimization as a bi-level MDP, one for the diffusion process and one for the environment MDP. Although the expressiveness of diffusion policies should be better than standard MLP policies, the authors found that DPPO typically underperforms MLP+PPO and attributed the phenomenon to the optimization difficulty of the bi-level MDP formulation. Building on this understanding, NCDPO formulates the diffusion process as a deterministic inference process by pre-determining the noises used during inference as conditions. In this way, the diffusion process is in fact *contracted* and therefore standard RL algorithms can be applied to optimize the policy. Since PPO still requires a stochastic policy, the authors also added a learnable noise to the final action, such that the probabilities of actions can be computed.   NCDPO is evaluated on a wide range of benchmarks, including Gym-Locomotion, Robomimic and a multi-agent task Google Research Football. Generally, NCDPO demonstrates superior performance and stable convergence compared to baseline methods.

**Questions:**

- Since you wish to formulate the generation process as deterministic inference, why not use ODE to solve the final actions or more straightforwardly, use flow policies?
- How are table 1 and table 2 related to the main text? I don't see any references to these tables.

**Ethical Concerns:**

["NO or VERY MINOR ethics concerns only"]

**Final Justification:**

Although the performance of the proposed algorithm still lags behind classic off-policy RL algorithms in term of sample efficiency, there exist scenarios where such PPO-based and diffusion-based algorithms may apply, as the authors showed in the rebuttal. Besides, NCDPO demonstrates adequate improvements over the baseline DPPO. Overall I would insist this is a borderline paper, but I am leaning towards acceptance since it is technically solid.

**Limitations:**

yes

**Quality:**

2

**Strengths And Weaknesses:**

- The presentation of this paper is clear and easy-to-follow.
- Some recent and relevant papers on diffusion policy optimization are missing from the related work section and comparison. To name a few:
    - QSM [1] and DAC [2], which optimize diffusion policies by aligning the score function with action gradients.
    - DIME [3], which fine-tunes diffusion policies with back-propagation though diffusion steps and seems to achieve better performance.
    - BDPO [4], which also formulates the diffusion policy optimization as a bi-level MDP but circumvents back-propagation through time.
    - SDAC [5], which uses re-weighted score matching to steer the generation towards high-Q regions.

- From the experiments on Gym-Locomotion, it seems both NCDPO and DPPO are not sample-efficient compared to standard RL algorithms and recent progresses in diffusion policy optimization. For example, SAC only requires 3M environment steps to achieve similar or superior performance that NCDPO needs 10M steps. Some other diffusion-based methods, like DIME and SDAC, only requires 1M steps. I would encourage the authors to expand the set of baseline methods and include 1) standard online RL algorithms (e.g. SAC, TD3); 2) offline-to-online methods and 3) diffusion-based methods (see below).

[1] Learning a Diffusion Model Policy from Rewards via Q-Score Matching

[2] Diffusion Actor-Critic: Formulating Constrained Policy Iteration as Diffusion Noise Regression for Offline Reinforcement Learning

[3] DIME:Diffusion-Based Maximum Entropy Reinforcement Learning.

[4] Behavior-Regularized Diffusion Policy Optimization for Offline Reinforcement Learning

[5] Efficient Online Reinforcement Learning for Diffusion Policy

---

> ### Author Rebuttal · Authors · 2025-07-31
>
> Thank you very much for your detailed review and valuable feedback. We sincerely appreciate the opportunity to address your comments and clarify our work. Below is our point-by-point response:
>
> ---
>
> ### 1. **Missing citations**
>
> Thank you for bringing these recent works to our attention. We will properly cite these related works in the revised version of our paper.
>
> ---
>
> ### 2. **Comparison with Additional Baselines**
>
> To highlight the strengths of our approach, we conducted additional experiments covering a diverse set of benchmarks:
> 1. **Standard RL approaches** including off-policy methods such as SAC and offline RL methods such as AWR and BCQ.
>
> 2. **Offline-to-online training with diffusion policy optimization approaches** including QSM, BDPO, and DAC.
>
> #### 2.1 **Comparison with Standard RL Approaches**
>
> We appreciate the reviewer's concerns on comparing NCDPO with standard RL approaches such as SAC. To fully show the advantage of NCDPO over standard RL methods, we conduct further experiments on a more challenging benchmark, **Franka Kitchen** benchmark. **DCDPO signifcantly performs the offline and off-policy RL baselines across all tasks.**
>
> | Task            | **NCDPO (Ours)** | DPPO | SAC | BC  | SAC-off | BEAR | BRAC-p | BRAC-v | AWR | BCQ | AlgaeDICE | COL |
> |-----------------|------------------|------|-----|-----|---------|------|--------|--------|-----|-----|-----------|-----|
> | kitchen-complete| **3.9**          | 3.58 | 0.0 | 1.4 | 0.6     | 0.0  | 0.0    | 0.0    | 0.0 | 0.3 | 0.0       | 1.8 |
> | kitchen-partial | **2.96**         | 2.68 | 0.0 | 1.4 | 0.0     | 0.5  | 0.0    | 0.0    | 0.6 | 0.8 | 0.0       | 1.9 |
> | kitchen-mixed   | **3.95**         | 2.9  | 0.0 | 1.9 | 0.1     | 1.9  | 0.0    | 0.0    | 0.4 | 0.3 | 0.1       | 2.0 |
>
> *(Maximum possible score: 4.0)* Results for addtional baselines are reported in \[1].
>
>
> #### 2.2 **Comparison with Diffusion Policy Optimization Baselines (Offline-to-Online)**
>
> We conducted additional experiments on the **Robomimic** benchmark, which is much more challenging than Gym Mujoco due to reward sparsity and long horizon.
>
> **Baselines for diffusion policy optimization in offline-to-online manner:** We include QSM, BDPO and DAC as diffusion policy optimization baselines. We first run offline training. Then we adapt these approches to online training setting by collecting environment interactions in a replay buffer and then perform model update on replay buffer.
>
> The table below shows that **NCDPO clearly outperforms diffusion policy optimization baselines**.
>
> | Task      | NCDPO (ours) | QSM  | BDPO | DAC |
> | --------- | ------------ | ---- | ---- | --- |
> | lift      | 100%         | 100% | 100% | 86% |
> | can       |**99.3%**     | 0%   | 84%  | 66% |
> | square    |**87.3%**     | 0%   | 0%   | 8%  |
> | transport |**96.7%**     | 0%   | 0%   | 0%  |
>
>
> ---
>
> ### 3. **ODE Solvers for Decoising Process**
>
> Thank you for this insightful question. To investigate the alternative of using an ODE solver as a deterministic generation process, we investigated the **DDIM**, a representative deterministic sampler for the denoising process. **In general, NCDPO outperforms DDIM+PPO, especially on the `Robomimic can` task**. We believe this is because **using stochastic denoising is better at exploration than deterministic denoising** and thus leads to better RL results.
>
> *Results comparing DDIM+PPO and NCDPO:*
>
> | Task               | NCDPO (ours) | DDIM+PPO |
> | ------------------ | ------------ | -------- |
> | hopper-medium      | 3297         | 3232     |
> | walker2d-medium    | 4248         | 4087     |
> | Robomimic lift     | 100%         | 97%      |
> | Robomimic can      | 99.3%        | 25%      |
>
>
> ---
>
> ### 4. **Missing reference to tables**
>
> Thank you for pointing this out. We will add the reference to Tables 1 and 2 at line 212 in the revised paper.
>
> ---
>
> We hope these responses help address your concerns. We are grateful for your thoughtful suggestions, and we hope you will consider reevaluating our work based on the additional analysis and clarifications provided.
>
> [1] D4RL: DATASETS FOR DEEP DATA-DRIVEN REINFORCEMENT LEARNING

---

> > ### Comment · Reviewer_ohxP · 2025-08-01
> >
> > I greatly appreciate the response of the authors. Some further comments and questions about the comparison studies in the rebuttal:
> >
> > Is the experiments in 2.1 fully offline? If so, the selected baseline algorithms are comparatively old and some of them are not even tailored for offline settings (such as SAC). Given that offline settings may incur additional challenges, why not stick to the original offline-to-online or online setting and include baselines such as SAC?
> >
> > On 2.2, could the authors explain what is the unique advantage of NCDPO that enables it to outperform other diffusion-based methods significantly?

---

> > > ### Author Response · Authors · 2025-08-03
> > >
> > > Thank you very much for your thoughtful and constructive feedback. We sincerely appreciate the opportunity to address your comments. Please find our detailed responses below:
> > >
> > > ### **Regarding Section 2.1**
> > >
> > > We appreciate your observation regarding the choice of baselines and the training settings. In the rebuttal table, we included two SAC baselines: one trained in a fully **online** setting (denoted as "SAC") and another in a fully **offline** setting (denoted as "SAC-off"). All other baselines, except for DPPO, were trained under fully offline conditions.
> > >
> > > We agree that training SAC in **offline-to-online** settings provides fairer comparisons. To address this, we have conducted new experiments using SAC trained in an offline-to-online manner. The updated results are summarized below:
> > >
> > > | Dataset  | NCDPO (Ours) | DPPO | SAC (Offline-to-Online) | SAC (Online) | SAC (Offline) |
> > > | -------- | ------------ |------| ----------------------- |-----|---------|
> > > | Complete | **3.90**         | 3.58 | 3.0                    | 0.0 | 0.6     |
> > > | Mixed    | **3.95**         | 2.68 | 2.4                    | 0.0 | 0.0     |
> > > | Partial  | **2.96**         | 2.9  | 1.9                    | 0.0 | 0.1     |
> > >
> > > As shown, NCDPO significantly outperforms SAC even when SAC is allowed to fine-tune from offline data, highlighting the robustness and effectiveness of our method. We also note that SAC demonstrated great instability during training.
> > >
> > >
> > > ### **Regarding Section 2.2**
> > >
> > > Thank you for your question regarding the unique advantages of NCDPO relative to other diffusion-based methods.
> > >
> > > Upon closer examination of these baselines, we find that they rely heavily on Q-function estimation. In complex tasks such as those in the Robomimic suite—characterized by long horizons and sparse rewards—learning accurate Q-functions becomes especially difficult, which in turn degrades the performance of these methods.
> > >
> > > In contrast, NCDPO builds upon PPO and is inherently less sensitive to inaccuracies in value estimation. By leveraging stable, on-policy optimization, NCDPO is able to perform reliably in settings where Q-function-based methods struggle. This contributes significantly to its strong performance in challenging manipulation tasks.

---

> > > > ### Comment · Reviewer_ohxP · 2025-08-06
> > > >
> > > > Thanks for the reply, which has adequately addressed most of my concerns. I have increased my score accordingly.

---

> ### Author Response · Authors · 2025-08-06
>
> Dear Reviewer ohxP:
>
> Thank you for taking the time to revisit our submission and adjust your final score. We sincerely appreciate the effort you invested in reading our rebuttal and reevaluating our work.

---

### Official Review · Reviewer_AZtn · 2025-07-03

**Clarity:** 1
**Significance:** 2
**Originality:** 3
**Rating:** 3
**Confidence:** 4

**Summary:**

This paper presents a novel and intuitive method to incorporate diffusion models into reinforcement learning (RL) by treating the diffusion process as a deterministic policy. The authors achieve this by augmenting the environment state with sampled noise, allowing the diffusion model to act like a deterministic function (similar to an MLP predicting the mean action). This reformulation enables the computation of action likelihoods and makes the approach compatible with standard policy gradient methods such as PPO. Experiments are conducted primarily on OpenAI locomotion tasks, comparing this method to several RL baselines suitable for diffusion architectures.

**Questions:**

**Areas to improve**

- **Improve the Writing and Formatting**:
The paper would benefit greatly from a clear and concise writing style, improved logical flow, and better-organized figures and captions.

- **Upgrade the Experimental Benchmark**:
Consider evaluating the method on more modern benchmarks. These would better showcase the potential of diffusion models and offer more meaningful insights.

- **Add Comparisons to MLP Policies**:
A key missing component is a comparison to a standard MLP policy (with and without state augmentation) using PPO. This would help clarify whether the proposed reparameterization of the diffusion model provides practical benefits.

- **Explore Multimodal or Visual Tasks**: Showing that your approach excels in environments requiring multimodal action generation or learning from complex observations would strongly reinforce your claims.

**Ethical Concerns:**

["NO or VERY MINOR ethics concerns only"]

**Final Justification:**

I am keeping my original score of weak reject as the experimental verification of the idea is not sufficient. In the rebuttal the authors ran a lot more experiments but that is (imo) beyond the scope of the rebuttal. In the rebuttal the authors also cannot provide new figures and videos which would be needed for a proper assessment.

**Limitations:**

yes

**Quality:**

2

**Strengths And Weaknesses:**

**Strengths**

- **Conceptual Simplicity and Innovation**: The core idea is simple yet creative. By augmenting the state with noise, the authors transform a stochastic diffusion model into a deterministic policy representation. This re-interpretation is elegant and avoids the need for complex inference or training pipelines that are common in previous RL + diffusion model works.

- **Practical Integration**: The proposed method integrates well with PPO without requiring significant architectural or algorithmic changes. This makes it practical and easy to adopt.


**Weaknesses**

1. **Writing and Presentation**
   - The writing lacks clarity in several parts and could benefit from more careful editing. The paper's storyline is at times hard to follow.
   - Figures are poorly formatted, and the overall typesetting does not meet the standard expected of top-tier conferences like NeurIPS.

2. **Limited and Outdated Experimental Setup**
   - The primary evaluation is conducted on the OpenAI locomotion benchmark, which is now considered outdated. These environments offer limited insight into the potential benefits of diffusion models, especially in 2025 when the field has largely moved toward more complex domains.
   - The main premise of diffusion models being beneficial in high-dimensional or multimodal tasks is not tested. There are no experiments involving image-based observations or complex sensory inputs.
   - The paper does not include a direct comparison between the proposed diffusion policy and a standard MLP policy (with and without noise state augmentation) under the same PPO setup. As such, it is unclear whether the diffusion model actually improves performance in these tasks.

3. **Lack of Demonstration of Unique Advantages**
   - The paper does not convincingly demonstrate a scenario where the proposed method is clearly better than existing approaches. All of these domains can be easily solved by an MLP and PPO. It would be particularly impactful to see the method succeed in tasks where standard policies struggle, such as those requiring multimodal output distributions or robust multi-step planning.

---

> ### Author Rebuttal · Authors · 2025-07-31
>
> Thank you very much for your thoughtful review and constructive feedback. We sincerely appreciate the opportunity to respond to your comments and to further clarify our work through additional experiments and analysis that highlight the advantages of our proposed method. Please find our point-by-point response below:
>
> ---
>
> ### **1. Writing and Presentation**
> We are grateful for your comments regarding the writing and presentation. In future revisions, we will carefully revise the paper to improve clarity, strengthen the logical flow of the narrative, and enhance the formatting and visual quality of all figures to meet the standards expected at top-tier venues.
>
> ---
>
> ### **2. Experiments on More Challenging Benchmarks**
> We fully agree that evaluating our method on more complex tasks would better demonstrate the strengths of our diffusion-based policy. To this end, we have conducted additional experiments on **Robomimic** and **Franka Kitchen**, both of which pose significant challenges due to **sparse rewards, long horizons, and multi-stage subtasks**. We compare NCDPO with DPPO and a set of additional offline RL methods.
>
> **Robomimic**
>
> Robomimic is a highly challenging benchmark.  Offline RL methods can not achieve high success rates on Robomimic, especially on `transport`.  Our method consistently outperforms DPPO and the additional baselines, with especially notable gains in the most difficult `transport` setting.
>
>
>
> | Task      | NCDPO (Ours) | DPPO  | IQL  | SAQ-IQL | CQL  | Robomimic CQL | SAQ-CQL | BC   | Robomimic BC | SAQ-BC |
> | --------- | ------------ | ----- | ---- | ------- | ---- | ------------- | ------- | ---- | ------------ | ------ |
> | lift      |   **100.0%**     | 99.7% | 58   | 90      | 64.2 | 92.7          | 90.8    | 59.5 | 100          | 90.1   |
> | can       |   **99.3%**      | 99.0% | 33.7 | 68      | 19.6 | 38            | 71.2    | 31.7 | 95.3         | 66.4   |
> | square    |   **87.3%**      | 87.0% | 26.9 | 46.7    | 0.0  | 5.3           | 44.3    | 19.3 | 78.7         | 45.3   |
> | transport |   **96.7%**      | 91.3% | 0.0  | 2.0     | 0.0  | 0.0           | 3.5     | 0.3  | 29.3         | 3.2    |
>
> Baseline results are taken from our paper and \[1].
>
> **Franka Kitchen**
>
> We also extended our evaluation to the **Franka Kitchen** environment using the D4RL dataset. This setting simulates real-world kitchen tasks with multiple, sequential goals under sparse rewards, presenting highly challenging tasks for RL training. Our method surpasses DPPO and all baselines across all subtasks:
>
> | Task             | NCDPO (Ours) | DPPO | SAC | BC  | SAC-off | BEAR | BRAC-p | BRAC-v | AWR | BCQ | AlgaeDICE | COL |
> | ---------------- | ------------ | ---- | --- | --- | ------- | ---- | ------ | ------ | --- | --- | --------- | --- |
> | kitchen-complete | **3.9**      | 3.58 | 0.0 | 1.4 | 0.6     | 0.0  | 0.0    | 0.0    | 0.0 | 0.3 | 0.0       | 1.8 |
> | kitchen-partial  | **2.96**     | 2.68 | 0.0 | 1.4 | 0.0     | 0.5  | 0.0    | 0.0    | 0.6 | 0.8 | 0.0       | 1.9 |
> | kitchen-mixed    | **3.95**     | 2.9  | 0.0 | 1.9 | 0.1     | 1.9  | 0.0    | 0.0    | 0.4 | 0.3 | 0.1       | 2.0 |
>
> Scores of offline RL baselines are reported in \[2].
>
> ---
>
> ### **3. Experiments on Visual Tasks (Robomimic)**
>
> To address the concern about image-based inputs, we have conducted experiments involving vision inputs the most difficult `square` and `transport` tasks. **NCDPO outperforms DPPO on visual tasks of Robomimic**. More results can be found in Appendix Section B.2.
>
> | Task (Image) | NCDPO (Ours) | DPPO  |
> | ------------ | ------------ | ----- |
> | square       | 84.3%        | 84.1% |
> | transport    | **96.0%**    | 92.5% |
>
>
> ---
>
> ### **4. Comparison with MLP Policies**
>
> We note that the fundamental advantage of NCDPO over MLP policies with standard RL (e.g. PPO) is that, **when demonstrations are available, diffusion policies are better at modeling diverse behaviors than MLP policies.** Here we provide a direct comparison between diffusion policy + NCDPO and MLP policy + PPO on three environments, *Google Research Football*, *Robomimic* and *Franka Kitchen*.
>
> **NCDPO consistently outperforms the MLP+PPO baseline across all tasks and scenarios**. This is because, on these three benchmarks, it is hard for a MLP policy to properly model the diverse behaviors in the demonstration dataset.
>
> **Google Research Football results:**
>
> | Scenario           | NCDPO (Ours) | MLP + PPO |
> | ------------------ | ------------ | --------- |
> | 3 vs 1 with Keeper | **87.4**     | 75.1      |
> | Counterattack Hard | **87.0**     | 80.0      |
> | Corner             | **78.3**     | 74.9      |
>
> **Robomimic results:**
>
> | Task      | NCDPO (Ours) | MLP + PPO |
> | --------- | ------------ | ---------- |
> | square    | **87.3%**    | 81.2%      |
> | transport | **96.7%**    |  0%        |
>
> **Franka Kitchen results:**
>
> | Dataset  | NCDPO (Ours) | MLP + PPO |
> | -------- | ------------ | ---- |
> | Complete | **3.9**      | 2.14 |
> | Mixed    | **3.95**     | 1.71 |
> | Partial  | **2.96**     | 1.22 |
>
>
> ---
>
> We will incorporate these clarifications, comparisons, and additional results into the main paper in future revisions.
>
> We hope this response has addressed the reviewer’s concerns and provided a clearer picture of the significance and applicability of our work. We would be sincerely grateful if the reviewer would kindly consider reassessing the evaluation of our submission in light of this additional information.
>
> [1] Action-Quantized Offline Reinforcement Learning for Robotic Skill Learning
> [2] D4RL: DATASETS FOR DEEP DATA-DRIVEN REINFORCEMENT LEARNING

---

> ### Author Response · Authors · 2025-08-04
>
> Dear Reviewer,
>
> Thank you for taking the time to review our work. We noticed that you submitted a final justification, but did not leave accompanying comments. We hope that our clarifications addressed your earlier feedback. Could you kindly confirm whether there are any remaining concerns that we could still address? If there is anything that remains unclear or unaddressed, we would greatly welcome further discussion or clarification. We truly appreciate your feedback and are happy to provide any additional information that may help.

---

### Official Review · Reviewer_2BZh · 2025-07-04

**Clarity:** 3
**Significance:** 3
**Originality:** 3
**Rating:** 5
**Confidence:** 3

**Summary:**

The paper proposes a novel algorithm combining the idea of PPO with diffusion models for the purpose of sequential decision-making. It considers each denoising step as a differentiable transformation to obtain an estimate of action likelihood, through out the denoising steps in diffusion process. The results are evaluated using standard offline RL datasets, showing solid improvement.

**Questions:**

- How does the algorithm compare with additional diffusion baselines such as Decision Diffuser, Decision Stacks, and CEIL?
- In addition to training RL on randomly initialized policies, can NCDPO further improve the performance of an already-trained RL policy?
- How does NCDPO compare with baselines on random datasets?

**Ethical Concerns:**

["NO or VERY MINOR ethics concerns only"]

**Final Justification:**

The authors addressed my concerns. I maintain the positive view of the paper.

**Limitations:**

Yes.

**Paper Formatting Concerns:**

None.

**Quality:**

3

**Strengths And Weaknesses:**

+ The idea of considering each denoising step as a differentiable transformation to obtain an estimate of action likelihood is novel. It leads to tractable optimization computing action likelihood estimates.
+ Evaluation shows that the proposed method outperforms baselines consistently in the evaluation. NCDPO outperforms baselines in both sample efficiency and final reward.
- The evaluation should add a few more baselines using diffusion policies, such as Decision Diffuser, Decision Stacks, and CEIL.
- Some theoretical analysis to demonstrate NCDPO's benefit in terms of sample efficiency would be helpful.
- In addition to training RL on randomly initialized policies, can NCDPO further improve the performance of an already-trained RL policy?

---

> ### Author Rebuttal · Authors · 2025-07-31
>
> Thank you very much for your detailed review and insightful feedback. We sincerely appreciate the opportunity to address your comments and clarify the contributions of our work. Please find our point-by-point response below:
>
> ---
>
> ### 1. **Comparison with Additional Baselines**
>
>
> **Comparison with Additional Baselines in Complex Environments:**
>
> Thank you for pointing out additional baselines. We make comparison with additional baselines including Decision Diffuser and offline RL methods. We present results on the **Franka Kitchen** environment, which is much more challenging than Gym Mujoco. As shown below, **our method NCDPO outperforms DPPO and additional baselines.** We note that offline RL methods struggle in this challenging environment and performs much worse than diffusion policy based approaches.
>
> | Task            | **NCDPO (Ours)** | DPPO | Decision Diffuser | SAC | BC  | SAC-off | BEAR | BRAC-p | BRAC-v | AWR | BCQ | AlgaeDICE | COL |
> |-----------------|------------------|------|-------------------|-----|-----|---------|------|--------|--------|-----|-----|-----------|-----|
> | kitchen-complete| **3.90**         | 3.58 | — (not reported)  | 0.0 | 1.4 | 0.6     | 0.0  | 0.0    | 0.0    | 0.0 | 0.3 | 0.0       | 1.8 |
> | kitchen-partial | **2.96**         | 2.68 | 2.28              | 0.0 | 1.4 | 0.0     | 0.5  | 0.0    | 0.0    | 0.6 | 0.8 | 0.0       | 1.9 |
> | kitchen-mixed   | **3.95**         | 2.90 | 2.60              | 0.0 | 1.9 | 0.1     | 1.9  | 0.0    | 0.0    | 0.4 | 0.3 | 0.1       | 2.0 |
>
> These results further highlight the effectiveness of our method in more complex settings. Scores for other baselines are reported in \[1].
>
>
> **Offline Methods are Orthogonal to Our Method:**
>
> We also kindly note that:
> - Offline methods are orthogonal to our focus. In this work, we focus on fine-tuning diffusion policies through online interaction with the environment.
> - Our method can also be seamlessly integrated with offline methods by using diffusion policies trained in offline manner as the starting point for online fine-tuning.
>
> ---
>
> ### 2. **Theoretical Analysis**
>
> We agree that formal guarantees would further strengthen our work, and we deeply considered this direction during our research.
>
> However, due to **lack of relevant works on backpropagating gradients through denoising steps in diffusion policies**, we found that a rigorous theoretical analysis exceeds the scope of this submission. Instead, we provide:
> - **Intuitive justification:** Although both NCDPO and DPPO fine-tune the denoising process with PPO, *the way they improve denoising steps is fundamentally different*. DPPO independently optimizes individual denoising step. In NCDPO, RL loss is computed only on the denoised action and intermediate denoising steps are improved by backpropagating the gradients through the denoising steps. **Our design simplifies the RL objective and jointly optimizes all denoising steps**, instead of treating them as individual decision-making steps.
> - **Empirical validation:** Our experiment results on training from randomly initialized policies confirm that NCDPO achieves better sample efficiency and final performance. We hypothesize that this is because **our design in NCDPO provides clearer and more informative gradients for denoising steps.**
>
> ---
>
> ### 3. **Fine-Tuning on Already Trained Policies**
>
> We kindly note that, in our main experiments in Sec 6.2 and 6.3, NCDPO is applied to fine-tune diffusion policies trained with behavior cloning on demonstration datasets. Our results show that **NCDPO can continously improve the performance of already trained diffusion policies with high sample efficiency.**
>
> ---
>
> ### 4. **Comparison with Baselines on Random Datasets**
>
> We appreciate the thoughtful question regarding the use of behavior cloning on random datasets. We would like to emphasize that since **our main focus is to investigate online fine-tuning for diffusion policies**, we select challenging settings where the datasets contain **structured and diverse demonstrations**. The challenging settings require diffusion policies to model diverse behaviors and thus are suitable to carry out our experiments.
>
> When using random datasets for behavior cloning, we believe this would lead to a weak initial policy and is similar to using a randomly initialized policy, as we discussed in Section 4.
>
> ---
>
> We thank the reviewer again for the constructive and thoughtful review. We hope our responses have addressed the concerns and helped clarify the contributions and scope of our work.
>
>
> **\[1] D4RL: Datasets for Deep Data-Driven Reinforcement Learning**

---

> > ### Comment · Reviewer_2BZh · 2025-08-05
> >
> > Thanks for providing the additional experiment results. That was my main concern, which is now addressed. I don't have further questions at this point.

---

### Official Review · Reviewer_tKab · 2025-07-20

**Clarity:** 3
**Significance:** 3
**Originality:** 3
**Rating:** 3
**Confidence:** 3

**Summary:**

This paper proposed a method to finetune diffusion policy using proximal policy optimization. The method is an improvement over DPPO[20] by using the noise as part of the states, so that the denoising process of the current and old diffusion policy share the same noise in finetuning. Experiments on RL and robotics benchmarks demonstrate the effectiveness of the proposed method

**Questions:**

It would be more convicing if the authors could report the results on robot hardwards as in [20]

**Ethical Concerns:**

["NO or VERY MINOR ethics concerns only"]

**Final Justification:**

I acknowledge that hardware experiments may not be strictly necessary if the proposed method demonstrated outstanding merits; however, this does not appear to be the case for this submission.

**Limitations:**

Yes

**Paper Formatting Concerns:**

No concerns

**Quality:**

3

**Strengths And Weaknesses:**

Strengths:
1. It is technical sound to use the same noise for \pi_\theta and \pi_\theta_old in finetuning to reduce loss variation and improve sample efficiency. It is somewhat novel and effective. The results are significantly better than [20] as reported in this paper
2. The proposed method is a straightforward modification from DDPO and can be easily adapted to different variants of diffusion policy finetuning methods
3. The paper reports the experimental results on multiple benchmark datasets including OpenAI Gym and Robomimics.

Weaknesses:
1. The novelty is limited. The proposed method is very similar to DDPO except that \pi_\theta and \pi_\theta_old share the same noise.
2. Experimental results are not convincing enough. All experiments are in simulation and there's no real world experiments.
3. The graphics quality and layout are not visually pleasing. The figure captions in Sec 6.4 is not informative. The paper seems to be completed in a rush
4. The title is uninformative. It provide little information about the proposed method and could just easily describe existing methods.

Overall the idea is technical sound and somewhat novel but the paper looks hastily written

---

> ### Author Rebuttal · Authors · 2025-07-31
>
> Thank you very much for your detailed review and valuable feedback. We sincerely appreciate the opportunity to address your comments and provide further clarification. Please find our point-by-point response below:
>
> 1. **Novelty and Similarity to DPPO:**
>    We respectfully believe there may be a misunderstanding regarding the distinction between our method and DPPO. Our approach is **fundamentally different** in the following ways:
>
>    * **DPPO** applies policy optimization independently to each denoising step.
>    * **In contrast, our method** optimizes only the *final* denoising step and backpropagates gradients through the *entire* diffusion process. This design yields **clearer and more informative gradients** for intermediate steps, leading to improved sample efficiency.
>
>    To the best of our knowledge, this is the **first PPO-based fine-tuning method** that introduces **backpropagation through diffusion timesteps**, which we consider the core innovation of our work.
>
> 2. **Lack of Real-World Experiments:**
>    We appreciate the importance of real-world validation and acknowledge this as a valuable direction for future work. However, **our focus is on algorithmic innovation**, specifically developing a more effective fine-tuning method for diffusion policies. *Sim-to-real transfer, while important, is beyond the scope of this work and orthogonal to our core contributions.* We believe our results demonstrate the value of the proposed method in simulated settings and provide a strong foundation for further deployment.
>
> 3. **Additional Results on Franka Kitchen Environment:**
>    To further validate the effectiveness of our approach, we conducted additional experiments on the **Franka Kitchen** environment. This setting simulates real-world kitchen with multiple sequential goals under sparse reward setting, presenting highly challenging tasks for RL training. We compare our approach with DDPO and offline RL methods.  **NCDPO consistently outperforms DPPO and additional baselines across all tasks**. The results are summarized below:
>
>    | Task             | **NCDPO (Ours)** | DPPO | SAC | BC  | SAC-off | BEAR | BRAC-p | BRAC-v | AWR | BCQ | AlgaeDICE | COL |
>    | ---------------- | ---------------- | ---- | --- | --- | ------- | ---- | ------ | ------ | --- | --- | --------- | --- |
>    | kitchen-complete | **3.90**         | 3.58 | 0.0 | 1.4 | 0.6     | 0.0  | 0.0    | 0.0    | 0.0 | 0.3 | 0.0       | 1.8 |
>    | kitchen-partial  | **2.96**         | 2.68 | 0.0 | 1.4 | 0.0     | 0.5  | 0.0    | 0.0    | 0.6 | 0.8 | 0.0       | 1.9 |
>    | kitchen-mixed    | **3.95**         | 2.90 | 0.0 | 1.9 | 0.1     | 1.9  | 0.0    | 0.0    | 0.4 | 0.3 | 0.1       | 2.0 |
>
> Scores for offline RL baselines are reported in \[1].
>
> 4. **Graphics Quality and Layout:**
>    Thank you for pointing out this issue. We will work to significantly enhance the visual quality, clarity, and layout of the figures in the revised version.
>
> 5. **Title is Uninformative:**
>    Thank you for this constructive suggestion. We will revise the title to **"Noise-Conditioned Diffusion Policy Optimization with Backpropagation Through Diffusion Timesteps"**, which we believe more accurately reflects the core ideas and contributions of our work.
>
> ---
>
> We hope our responses have addressed the concerns raised and clarified the novelty and impact of our contributions. We sincerely hope you will consider re-evaluating our work in light of these clarifications.
>
> **\[1] D4RL: Datasets for Deep Data-Driven Reinforcement Learning**

---

> > ### Comment · Reviewer_tKab · 2025-08-07
> >
> > Thanks for the response. Backpropagation through time is technical sound and seems to be effective

---

> > > ### Author Response · Authors · 2025-08-07
> > >
> > > Dear Reviewer tKab,
> > >
> > > Thank you very much for your thoughtful and constructive review, as well as for recognizing the technical soundness of our proposed method.
> > >
> > > We have carefully addressed the specific concerns you raised in our rebuttal. Given that your overall score remains at 3 (borderline reject), we would be sincerely grateful if you could kindly elaborate on any remaining factors that led to this assessment.
> > >
> > > If all your concerns have been satisfactorily addressed, we would greatly appreciate your support in improving the overall evaluation of our submission, as your endorsement is very important to us.

---

> > > > ### Comment · Reviewer_tKab · 2025-08-08
> > > >
> > > > My other concerns are: (1) low-quality presentation, (2) better than my initial understanding but still limited novelty, and (3) Although I acknowledge this is not strictly required for acceptance, it would be stronger if there are hardware experiments. I maintain my original rating and leave the decision to the AC

---

> ### Author Response · Authors · 2025-08-06
>
> Dear Reviewer tKab,
>
> Thank you for your thoughtful review. We appreciate the concerns you raised and have provided detailed responses in our rebuttal.
>
> We would be grateful to know whether our clarifications have resolved those points. If anything remains unclear, or if further details would be helpful, we sincerely welcome further discussion. Your feedback is very valuable.

---

> ### Author Response · Authors · 2025-08-08
>
> Dear Reviewer tKab,
>
> Thank you very much for your thoughtful and constructive review. Please find our point-by-point response below:
>
> 1. **Low Presentation Quality**:
>    We appreciate you highlighting this concern. Due to the OpenReview policy, we are unable to update the paper during the rebuttal and discussion phases. However, we assure you that we will make substantial improvements to the organization and presentation of the paper in future revisions.
>
> 2. **Limited Novelty**:
>    We would appreciate it if you could specify the grounds for your assessment of “limited novelty.” In particular, could you indicate which aspects of our contribution you believe lack novelty? Our core contribution lies in introducing the backpropagation of policy loss through diffusion timesteps during PPO optimization—an approach that, to the best of our knowledge, has not been previously explored and constitutes a significant methodological advancement. We would be glad to provide further clarification or supporting details if needed.
>
> 3. **Hardware Experiments**:
>    Thank you for your comment. Based on your comment, we understand that the absence of hardware experiments does not affect the paper’s acceptance.
>
> We are sincerely thankful for the constructive suggestions you have provided. However, we respectfully disagree with the assessment of “limited novelty.” We would be happy to provide further clarification if the reviewer could indicate the specific reasons or aspects on which this assessment is based.

---

### Decision · Program_Chairs · 2025-09-17

**Decision:**

Reject

**Comment:**

This paper reformulates diffusion policies as noise-conditioned deterministic policies and enables PPO gradients to be backpropagated through the denoising steps. The idea is technically sound and addresses certain limitations of DPPO. Empirically, NCDPO, the proposed method, shows consistent improvements over DPPO across several domains, including more challenging benchmarks such as RoboMimic and Franka Kitchen. The rebuttal added further experiments that helped clarify the potential advantages of the approach.

The main concerns on NCDPO are about novelty relative to recent diffusion RL methods, reliance on older benchmarks, and presentation quality, with writing and figures below NeurIPS standards. The absence of real-world experiments was noted, but not considered critical.

Another concern is that the rebuttal introduced additional results that play an important role in strengthening the case for acceptance. While these findings are promising, their significance would ideally warrant a more thorough re-review. This consideration contributes to the view that the paper remains slightly below the bar for acceptance at this time.